# MCM: Multi-layer Concept Map for Efficient Concept Learning from Masked Images

## Abstract

Masking strategies commonly employed in natural language processing are still underexplored in vision tasks such as concept learning, where conventional methods typically rely on full images. However, using masked images diversifies perceptual inputs, potentially offering significant advantages in concept learning with large-scale Transformer models. To this end, we propose Multi-layer Concept Map (MCM), the first work to devise an efficient concept learning method based on masked images. In particular, we introduce an asymmetric concept learning architecture by establishing correlations between different encoder and decoder layers, updating concept tokens using backward gradients from reconstruction tasks. The learned concept tokens at various levels of granularity help either reconstruct the masked image patches by filling in gaps or guide the reconstruction results in a direction that reflects specific concepts. Moreover, we present both quantitative and qualitative results across a wide range of metrics, demonstrating that MCM significantly reduces computational costs by training on fewer than 75% of the total image patches while enhancing concept prediction performance. Additionally, editing specific concept tokens in the latent space enables targeted image generation from masked images, aligning both the visible contextual patches and the provided concepts. By further adjusting the testing time mask ratio, we could produce a range of reconstructions that blend the visible patches with the provided concepts, proportional to the chosen ratios.

## 1 Introduction

Humans often learn concepts through contextual understanding by recognizing relationships among features. Similarly, in a reconstruction task, masking a large portion of the input enables the model to leverage context from unmasked regions, thereby potentially enhancing the learning of dependencies that define concepts. By deprioritizing pixel-level details, the masking strategy encourages the focus on consistent features across instances, leading to better generalization. While masking strategies are well-studied in language tasks, they still remain underexplored in vision tasks, particularly in the context of concept learning, where existing studies typically focus on learning from full images. Consequently, we aim to investigate whether the masking objective diversifies perceptual inputs and could provide additional benefits for concept learning with large-scale Transformer models.

We propose the Multi-layer Concept Map (MCM) method to facilitate masked concept learning through vision reconstruction tasks. Specifically, we leverage cross-attention for learning concept tokens at various granularity levels from masked images. These concept tokens assist in reconstructing input images by filling gaps or guiding reconstruction results in a specific direction for effective concept manipulation. Our method employs an asymmetric concept learning architecture, establishing correlations between different encoder and decoder layers (Figure 1). This architecture allows concept tokens to be updated using backward gradients from reconstruction tasks, enabling decoder layers to focus on distinct encoder layer outputs and enhancing reconstruction performance.

MCM is an efficient method for masked concept learning with significantly reduced computational cost, achieved by masking large portions of image patches and using an asymmetric model architecture. Nevertheless, extensive experimental results demonstrate that MCM could also enhance concept prediction performance compared to conventional methods. Furthermore, even with extremely limited input information, the model effectively learns a set of concepts that guide reconstruction

results in a specific direction for concept manipulation. The reconstructed images align with the visible unmasked image tokens while reflecting the provided concept tokens. Consequently, MCM learns effective concept tokens by training on less than 75% of the total image patches, achieving enhanced or competitive performance in both prediction and reconstruction tasks.

Overall, our main contributions are three-fold:

(1) We propose the Multi-layer Concept Map (MCM) method to facilitate masked concept learning through vision reconstruction tasks, which involves the masked concept encoder and multi-layer concept mapping architecture.

(2) This study investigates two dedicated loss functions to enhance the model's ability in concept prediction, especially for training on unbalanced concept classes, i.e., the disentanglement loss and weighted concept loss.

(3) The extensive quantitative and qualitative analysis involves concept prediction performance, reconstruction quality measured by Fréchet Inception Distance, computational cost, and diverse visualizations. MCM learns effective concept tokens using less than 75% of image patches while achieving competitive performance.

The remainder of this paper is structured as follows. Section 2 reviews the most recent work on masked image reconstruction and disentangled representation learning. Section 3 demonstrates the essential definitions, assumptions, and technical underpinnings of the proposed method. Section 4 presents a thorough examination using a broad range of metrics to assess concept prediction and reconstruction performance. Section 5 concludes our findings and gives out future directions.

## 2 RELATED WORK

### 2.1 MASKED IMAGE RECONSTRUCTION

Masked image modeling has emerged as a pivotal learning technique in computer vision He et al. (2022); Yue et al. (2023); Zheng et al. (2023); Chen et al. (2023); Fu et al. (2024). For instance, Masked Autoencoders He et al. (2022) learn to reconstruct missing patches given only a small subset of visible patches reducing computational cost. Masked Diffusion Transformer Zheng et al. (2023); Gao et al. (2023) demonstrates enhanced training efficiency and generation results using a denoising diffusion objective on masked images. Cross-Attention Masked Autoencoders Fu et al. (2024) used cross-attention in the decoder to query visible tokens for masked image patch reconstruction. The cross-attention component takes a weighted sum of the visible tokens across different input blocks to fuse the features for each decoder block, leveraging low-level information for reconstruction.

### 2.2 DISENTANGLED REPRESENTATION LEARNING

Disentangled concept learning Bengio et al. (2013); Higgins et al. (2017); Locatello et al. (2019); Härkönen et al. (2020); Anirudh et al. (2021); Yang et al. (2022); Sun et al. (2023); Ismail et al. (2023) aims to uncover the underlying explanatory factors hidden within observed data. For example, methods such as $\beta$-VAE Higgins et al. (2017) and FactorVAE Kim & Mnih (2018) search for directions in the latent space that correlate with distinct human-interpretable concepts. Moreover, Concept Tokenization Yang et al. (2022) focuses on learning disentangled object representations and inspecting latent traversals for various factors. Additionally, Concept Bottleneck models Ismail et al. (2023); Yuksekgonul et al. (2022); Oikarinen et al. (2023); Yang et al. (2023) learn representations that correspond to specific human-understandable concepts. Energy-based methods Du et al. (2020); Li et al. (2022); Du et al. (2023) aim to compute energy functions of various concepts and combine their probability distributions achieving conjunctions, disjunctions, and negations of various concepts.

Conventional concept learning methods typically rely on fully observable images for training. While masking strategies have proven effective in reducing computational cost in natural language processing, their usage in concept learning tasks remains underexplored. This is primarily because masking a large portion of image patches greatly limits the information available for disentangling effective concepts. To address this challenge, we integrate learnable concept tokens at various granularity levels into the masked reconstruction process using the asymmetric Multi-layer Concept Map (MCM)

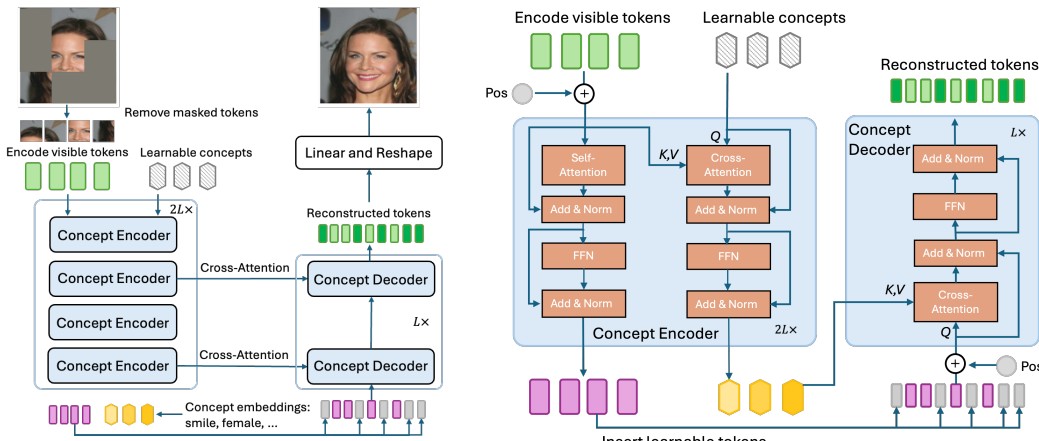

(a) MCM randomly masks an image. A set of learnable concepts is learned at each encoder layer alongside the visible tokens. Learnable mask tokens initialized with Gaussian noise are added at the masked positions in the encoder output. The decoder then utilizes the mask tokens for reconstruction via cross-attention, leveraging concept tokens at various granularities. For computational efficiency, in the asymmetric architecture, concept tokens from every two encoder layers are used.

(b) In the encoder layer, self-attention and a feedforward network (FFN) process the input visible tokens, while cross-attention updates concept tokens using the input tokens, followed by an FFN. Skip connections and layer normalization are utilized throughout. In the decoder layer, cross-attention updates mask tokens using concept tokens learned from specific encoder layers as keys and values. The decoder layer also employs an FFN, skip connections, and layer normalization.

Figure 1: (a) The architecture of the proposed Multi-layer Concept Map (MCM) method. (b) The detailed framework of the encoder and decoder layers.

architecture. This approach could not only reduce computational cost for learning effective concepts but also enhance model's concept prediction capability. Our goal is to advance the masked concept learning objective, paving the way for more efficient model architectures.

## 3 METHOD

In this section, we introduce the Multi-layer Concept Map (MCM) method, which involves the masked concept encoder and multi-layer concept decoder architecture. In addition to the reconstruction target, we devise two dedicated loss functions to enhance the model's concept prediction, especially for unbalanced concept classes, i.e., the disentanglement loss and weighted concept loss.

### 3.1 CONCEPT ENCODING FROM MASKED IMAGES

MCM employs multiple encoder layers to encode visible tokens and learn concept tokens at various granularity levels. Then, decoder layers aim to reconstruct the masked patches using the concept tokens and contextual information from the visible tokens. In particular, MCM divides images into patches and processes them using attention mechanisms Vaswani et al. (2017). While model architectures such as convolution layers could also be used for encoding and decoding, the masking strategy is typically utilized in Transformer models.

Let $x \in \mathbb{R}^{H \times W \times C}$ be an input image, where $(H, W)$ represents the resolution of the image and $C$ is the number of channels. The image $x$ is partitioned into a sequence of patches $x_p \in \mathbb{R}^{N \times (P^2 \cdot C)}$, where $(P, P)$ denotes the resolution of each patch, and $N = \frac{HW}{P^2}$ is the total number of patches. These patches are then mapped to embeddings $v_p \in \mathbb{R}^{N \times E}$ via a linear projection $W^P \in \mathbb{R}^{(P^2 \cdot C) \times E}$. With a mask ratio $r$, we randomly remove $\lfloor rN \rfloor$ tokens from the input, leaving only $N - \lfloor rN \rfloor$ visible tokens as input $v_{\text{masked}} \in \mathbb{R}^{(N-\lfloor rN \rfloor) \times E}$ to the model. A higher mask ratio enhances computational efficiency but reduces contextual information for concept learning. Therefore, an optimal mask ratio likely exists, balancing both efficiency and performance.

An encoder model that consists of multiple attention layers takes the encoded visible tokens $v_{\text{masked}} \in \mathbb{R}^{(N-\lfloor rN \rfloor) \times E}$ and a set of learnable concept tokens $C_0 \in \mathbb{R}^{M \times E}$ as the input (Figure 1). Notably, we initialize the concept tokens using Gaussian noise and share them *across batch samples*. Then, for each encoder layer $l_{\text{encoder}}$, we employ multi-head cross-attention to update the concept tokens $C_{l_{\text{encoder}}} \in \mathbb{R}^{M \times E}$ using the visible patch tokens $v_{\text{masked}}^{l_{\text{encoder}}}$ as the key and value. As a result, the output of an attention head $i$ is a weighted sum of the values, i.e., $\text{softmax} \left( \frac{W_i^Q C_{l_{\text{encoder}}} (W_i^K v_{\text{masked}}^{l_{\text{encoder}}})^T}{\sqrt{E}} \right) W_i^V v_{\text{masked}}^{l_{\text{encoder}}}$, where $W^Q$, $W^K$, and $W^V$ are the projection weights for the query, key, and value, respectively. Finally, we use a feedforward network to obtain $C_{l_{\text{encoder}}+1}$. Note that we do not employ self-attention for the learned concept tokens, which enables individual concept updates thus diversifying concept tokens. Moreover, we process the visible patch tokens $v_{\text{masked}}^{l_{\text{encoder}}+1}$ for extracting high-level contextual information. In particular, we employ self-attention followed by a feedforward network to learn associations among visible tokens. For learning both concept and visible patch tokens, the skip connection and layer normalization is employed throughout. All the feedforward networks consist of two linear layers with a GELU activation Hendrycks & Gimpel (2016) in between. We stack multiple concept encoder layers $l_{\text{encoder}} \in \{1, 2, \ldots, L_{\text{encoder}}\}$ in depth to obtain the latent representations of concept tokens $C_{L_{\text{encoder}}}$ and visible tokens $v_{\text{masked}}^{L_{\text{encoder}}}$ as the outputs of the encoder model.

## 3.2 IMAGE DECODING WITH MULTI-LAYER CONCEPT MAPPING

To reconstruct the masked patches, we add learnable mask tokens $v_{\text{init}} \in \mathbb{R}^{\lfloor N \times r \rfloor \times E}$ at the positions of masked patches in the encoder output. These mask tokens are initialized with values drawn from a Gaussian distribution. Notably, we concatenate and rearrange the visible tokens $v_{\text{masked}}^{L_{\text{encoder}}} \in \mathbb{R}^{(N-\lfloor N \times r \rfloor) \times E}$ and the mask tokens $v_{\text{init}} \in \mathbb{R}^{\lfloor N \times r \rfloor \times E}$ based on the mask indices $Z \in \mathbb{R}^{\lfloor N \times \gamma \rfloor}$, as the decoder input $v_{\text{full}}^0 \in \mathbb{R}^{N \times E}$. Moreover, the decoder model computes on the full $N$ image tokens that are much more than the $N - \lfloor N \times r \rfloor$ visible tokens processed by the encoder model. To alleviate the computational cost induced by the decoding process with the full patch length, we employ an asymmetric architecture for the decoder using half the number of layers as the encoder.

A specific multi-layer concept mapping architecture is devised based on cross-attention components between paired encoder and decoder layers. This enables the reconstruction of mask tokens using the learned concept tokens from various encoder layers. With the asymmetric architecture, every two encoder layer's concept tokens are utilized for reconstructing the mask tokens of a specific decoder layer through cross-attention MHA($\cdot$), using the concept tokens as the key and value, i.e., $\hat{v}_{\text{full}}^{l_{\text{decoder}}} \leftarrow \text{MHA}(v_{\text{full}}^{l_{\text{decoder}}}, C_{L_{\text{encoder}}-2l_{\text{decoder}}})$. Then, a feedforward network FF($\cdot$) computes the decoder layer output $v_{\text{full}}^{l_{\text{decoder}}+1} \leftarrow \text{FF}(\hat{v}_{\text{full}}^{l_{\text{decoder}}})$. Note that we refrain from using self-attention in the decoder model to prevent the model from overly focusing on contextual visible tokens, allowing the model to prioritize the concept token learning. Consequently, after stacking $L_{\text{decoder}}$ decoder layers, the full tokens $v_{\text{full}}^{L_{\text{decoder}}}$ are converted into a pixel-level image $\hat{x} \in \mathbb{R}^{N \times (P^2 \times 3)}$ as the reconstruction result.

### 3.2.1 MASKED RECONSTRUCTION LOSS

MCM updates learnable concept tokens through a reconstruction objective. In particular, we compute the reconstruction loss between the decoder output $\hat{x}$ and the input image $x$ using the mean squared error loss: $\ell_{\text{MSE}}(X, C_0) = \frac{1}{B} \sum_{i=1}^{B} (\hat{x}_i - x_i)^2$. To enhance computational efficiency, we specifically compute the loss for the mask tokens as follows:

$$\ell_{\text{re}}(X, C_0) = \frac{\sum_{j=1}^{N} \ell_{\text{MSE}}^j \cdot \mathbb{1}[j \in Z]}{\lfloor N \times \gamma \rfloor},$$

where $Z$ is the indices of mask tokens, $\mathbb{1}[j \in Z]$ is an indicator function that outputs 1 if the $j$th token is a masked position and 0 otherwise, and $\ell_{\text{MSE}}^j$ is the mean squared error loss for the $j$th token.

### 3.2.2 DISENTANGLEMENT LOSS

We aim to encourage the model to learn mutually exclusive representations for various concept tokens, thus enhancing its generalization to unseen test samples. We devise a disentanglement loss

by randomly swapping a concept in the latent space with its antonym and identifying the modified concepts from reconstruction results. In particular, given an image $x$ and its predicted concepts $C_{L_{\text{encoder}}}$ in the latent space, we select a specific concept position $j \in \{1, 2, ..., M\}$ based on a random binary mask $U \in \{0, 1\}^M$, where exactly one position has a value of 1, indicating where the concept modification occurs. We then replace the concept $C_{L_{\text{encoder}}}^j$ with its antonym token $\mathcal{O}(C_{L_{\text{encoder}}}^j)$, where $\mathcal{O}(\cdot)$ is a function that maps a concept to its antonym. The decoder $f_{\text{decoder}}$ then reconstructs an image $\tilde{x}$ with the modified concepts $\hat{C}_{L_{\text{encoder}}}$. Intuitively, if the model learns the differences among concepts, the predicted concepts in the reconstruction results would show modifications only in the selected one. To verify this, we input a reconstruction result $\tilde{x}$ into the encoder $f_{\text{encoder}}$ to obtain the predicted concepts $\tilde{C}_{L_{\text{encoder}}}$, which are expected to match the modified concepts $\hat{C}_{L_{\text{encoder}}}$. Consequently, we devise the disentanglement loss as follows:

$$\hat{c}_{L_{\text{encoder}}}^i = \{U_j \cdot \mathcal{O}(c_{L_{\text{encoder}}}^{i,j}) + (1 - U_j) \cdot c_{L_{\text{encoder}}}^{i,j}\}_{j=1}^M,$$

$$\tilde{c}_{L_{\text{encoder}}}^i = f_{\text{encoder}} f_{\text{decoder}}(\hat{c}_{L_{\text{encoder}}}^i),$$

$$\ell_{\text{disentangle}}(X, C_0, U) = \frac{1}{B} \sum_{i=1}^B \left(\hat{c}_{L_{\text{encoder}}}^i - \tilde{c}_{L_{\text{encoder}}}^i\right)^2.$$

### 3.2.3 WEIGHTED CONCEPT LOSS

Concepts involved in a dataset are often biased. To encourage the model to focus more on underrepresented concepts during training, we further propose the weighted concept loss to adjust the impact of each concept's prediction error using the concept's frequency in a batch. In the latent space, we utilize a set of concept embeddings $C_{\text{prototype}}$ learned through approaches such as self-supervised learning, where numerical concept labels are converted to semantic embeddings of specific concepts, each with a dimension $E$. Notably, given a batch of size $B$ and the predicted concept tokens $c_{L_{\text{encoder}}}^{i,j}$, for each sample $i$ and concept index $j$, the loss is formulated as:

$$\ell_{\text{concept}} = \frac{1}{B} \sum_{i=1}^B \sum_{j=1}^M w_{i,j} \cdot \left(c_{L_{\text{encoder}}}^{i,j} - c_{\text{prototype}}^{i,j}\right)^2,$$

where $w_{i,j}$ represents a weight assigned to the error of the concept $j$ in sample $i$. These weights are inversely proportional to the frequency values of the concepts in the batch: $w_{i,j} = \frac{S}{\text{freq}(c_{\text{prototype}}^{i,j}) + \epsilon}$, where $\text{freq}(c_{\text{prototype}}^{i,j})$ is the frequency of the concept $c_{\text{prototype}}^{i,j}$ in the batch $\{\{c_{\text{prototype}}^{i,j}\}_{j=1}^M\}_{i=1}^B$, $S$ is a scaling constant to control the magnitude of the weights, and $\epsilon$ is a small constant added to avoid division by zero and to smooth the weights. The weighted concept loss gives higher importance to less frequent concepts by increasing their corresponding weights. Conversely, more frequent concepts have lower weights with reduced contribution. Additionally, we use coefficients $\alpha$ and $\beta$ to balance the various loss components, i.e., $\mathcal{L}(X, C_0, C_{\text{prototype}}, U) = \ell_{\text{re}} + \alpha \cdot \ell_{\text{disentangle}} + \beta \cdot \ell_{\text{concept}}$.

## 4 EXPERIMENTS

This section describes the detailed experimental settings. We present both quantitative and qualitative results on concept prediction performance, reconstruction quality, and computational cost, followed by extensive ablation studies. We demonstrate the method's image editing capability by showing how specific concept features could be disentangled exclusively from input images and how multiple concepts in the latent space could be combined. The results indicate that using an optimized mask ratio not only reduces computational cost but enhances model concept learning performance.

### 4.1 SETTINGS

#### 4.1.1 DATASETS

We conduct our experiments on the CelebA dataset Liu et al. (2015), a large-scale collection of celebrity face images annotated with rich binary attributes that capture diverse factors of variation. From this dataset, we select a subset of 11 semantically meaningful concepts exhibiting different

Table 1: Performance metrics at different mask ratios for small, base, and large-sized models. Note that since higher input image resolutions generally yield higher FID scores for larger models, we typically compare FID scores given the same image complexity.

| Size | Mask Ratio | Accuracy↑ | Precision↑ | Recall↑ | F1-Score↑ | FID↓ | Training T (h)↓ |
|------|-----------|-----------|-----------|---------|-----------|------|-----------------|
| Small | 0.0 | 0.877 | 0.514 | 0.563 | 0.537 | **1.053** | 4.13 |
| | 0.1 | 0.878 | 0.518 | 0.566 | 0.541 | 1.263 | 4.08 |
| | 0.25 | 0.882 | 0.521 | 0.57 | 0.544 | 1.438 | 4.03 |
| | **0.5** | **0.884** | **0.523** | **0.572** | **0.546** | 2.045 | 3.87 |
| | 0.75 | 0.854 | 0.496 | 0.542 | 0.518 | 4.655 | 3.76 |
| | 0.9 | 0.845 | 0.487 | 0.531 | 0.508 | 10.97 | **3.57** |
| Base | 0.0 | 0.935 | 0.857 | 0.815 | 0.835 | 1.972 | 6.45 |
| | 0.1 | 0.943 | 0.876 | 0.828 | 0.851 | 1.886 | 6.33 |
| | **0.25** | **0.945** | 0.881 | **0.839** | **0.860** | **1.884** | 6.12 |
| | 0.5 | 0.942 | 0.888 | 0.812 | 0.849 | 2.67 | 6.0 |
| | 0.75 | 0.935 | **0.901** | 0.786 | 0.839 | 5.386 | 5.79 |
| | 0.9 | 0.917 | 0.847 | 0.729 | 0.784 | 14.279 | **5.51** |
| Large | 0.0 | 0.946 | 0.807 | 0.785 | 0.796 | 4.195 | 9.82 |
| | 0.1 | 0.925 | 0.753 | 0.74 | 0.746 | 4.033 | 9.76 |
| | **0.25** | **0.955** | **0.821** | **0.796** | **0.808** | 4.027 | 9.65 |
| | 0.5 | 0.937 | 0.739 | 0.714 | 0.726 | 5.871 | 8.93 |
| | 0.75 | 0.913 | 0.652 | 0.674 | 0.663 | 7.755 | 8.23 |
| | 0.9 | 0.835 | 0.528 | 0.533 | 0.53 | 12.993 | **8.11** |

Table 2: Model performance comparison using base-sized models. The results show that MCM achieves superior concept-learning performance, while other baselines augmented with the concept-learning objective either underperform in accuracy or suffer from degraded reconstruction quality.

| Method | Accuracy↑ | Precision↑ | Recall↑ | F1-Score↑ | FID↓ | Size (M) |
|--------|-----------|-----------|---------|-----------|------|----------|
| MCM | 0.945 | 0.881 | 0.839 | 0.860 | 1.884 | 112.2 |
| MCM + Weighted and Disentangle Losses | **0.946** | **0.886** | **0.859** | **0.872** | 1.605 | |
| Masked Autoencoder He et al. (2022) | 0.537 | 0.809 | 0.750 | 0.778 | **1.274** | 114.8 |
| Masked DiT v2 Gao et al. (2023) | 0.735 | 0.189 | 0.241 | 0.146 | 32.38 | 87.0 |
| Cross-Attention MAE Fu et al. (2024) | 0.703 | 0.437 | 0.673 | 0.469 | 3.295 | 53.2 |
| Concept Tokenization Yang et al. (2022) | 0.925 | 0.756 | 0.686 | 0.719 | 3.971 | 118.7 |
| CLIP-Based MAE He et al. (2022) | 0.908 | 0.788 | 0.731 | 0.759 | 6.059 | 94.5 |

levels of frequency and visual distinctiveness (please refer to Appendix A.2 for distribution details). To enable systematic evaluation across positive and negative attributes, we construct antonymic counterparts by prefixing each concept with "Not" (e.g., Smiling vs. Not Smiling). This design enables performance evaluation under both frequent and underrepresented concepts, while ensuring a balanced treatment of concept presence and absence.

### 4.1.2 MODELS

Models of varying sizes (Small, Base, and Large) were trained with hyperparameters tailored to their complexity. For the detailed architecture settings, please refer to Appendix A.1. To learn concept tokens that contribute not only to concept prediction but to masked image reconstruction tasks, we convert binary concept labels into 512-dimensional embeddings using a pretrained CLIP model Radford et al. (2021). These concept embeddings provide guidance in the latent space, facilitating the reconstruction of masked patches. All experiments were performed using four A100 GPUs.

### 4.2 QUANTITATIVE RESULTS

### 4.2.1 CONCEPT PREDICTION AND RECONSTRUCTION PERFORMANCE

We aim to study how various training mask ratios affect the model's performance in the concept prediction and reconstruction tasks. We measure numerical prediction performance based on ac-

Table 3: Ablation studies on the various components of MCM.

| Method | Accuracy↑ | Precision↑ | Recall↑ | F1-Score↑ | FID↓ |
|---|---|---|---|---|---|
| Default MCM | 0.945 | 0.881 | 0.839 | 0.860 | 1.884 |
| W/ Weighted Loss | **0.946** | 0.886 | 0.852 | 0.869 | 1.734 |
| W/ Disentanglement Loss | 0.945 | **0.893** | 0.841 | 0.866 | **1.584** |
| W/ Weighted and Disentanglement Losses | **0.946** | 0.886 | **0.859** | **0.872** | 1.605 |
| W/O Branches | 0.925 | 0.756 | 0.686 | 0.719 | 3.971 |
| W/O Learnable Latent Concepts | 0.944 | 0.831 | 0.781 | 0.805 | 2.535 |
| Repetitive Latent Concepts | 0.926 | 0.753 | 0.706 | 0.729 | 2.167 |

curacy, precision, recall, and F1 score, evaluating reconstruction performance based on the Fréchet Inception Distance (FID) score. Table 1 demonstrates the impact of varying training mask ratios, showing an optimized mask ratio of 0.5 for the small-sized model and 0.25 for the base and large-sized models. For each entry, we trained the small and base-sized models for 500 epochs and the large-sized model for 100 epochs. In particular, for the base and large-sized models, as the mask ratio increases beyond 0.25, accuracy and F1-score start to decline, and FID increases substantially.

Table 2 demonstrates the comparison with baseline methods including the Masked Autoencoder (MAE) He et al. (2022), the Masked Diffusion Transformer v2 (Masked DiT v2) Gao et al. (2023), the Cross-Attention MAE Fu et al. (2024), the Concept Tokenization method Yang et al. (2022), and the CLIP-Based MAE He et al. (2022). For the CLIP-Based MAE model, we employ concept embeddings from the pretrained CLIP model to supervise representation learning in the MAE's latent space. For the other methods, we follow the architecture designs and hyperparameter settings of the base-sized model. For concept prediction, we leverage the latent representations of the unmasked visible tokens as input to an MLP attached to the model, with binary concept labels from the CelebA dataset at each

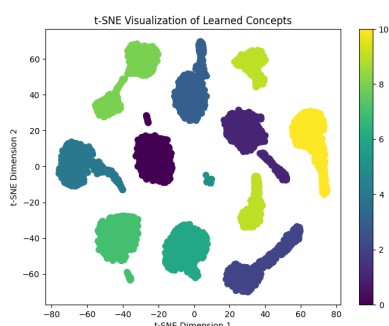

Figure 2: The t-SNE visualization of learned concept tokens in the latent space of MCM.

concept position. All models are trained from scratch. Note that our method (MCM) does not require binary concept labels for training. Instead, it learns a set of specific concept tokens directly in the latent space, as illustrated in Figure 2.

The comparison demonstrates that, even with supervised guidance from binary labels, MAE falls short of achieving concept-learning performance comparable to the proposed MCM method. The advantages of the proposed MCM are evident not only in its superior concept prediction accuracy but also in its novel image editing capabilities for masked image reconstruction, which is a functionality that MAE cannot provide. Additional comparison of the reconstruction results are presented in Figure 6. Moreover, inducing the concept-learning objective into other methods, such as Masked DiT v2, substantially degrades their masked reconstruction ability, as reflected in lower FID scores. In comparison to MCM, these methods neither attain competitive concept-learning performance nor support editing of masked images.

### 4.2.2 ABLATION STUDIES

We conducted extensive ablation studies to evaluate the benefits of the various components in the proposed MCM method. In addition to the ablations for the proposed two types of losses, i.e., the weighted concept loss and disentanglement loss, we specifically consider the following ablations:

(1) **W/O Branches:** Instead of using learned concept tokens from different encoder layers with the cross-attention mechanism, self-attention is employed to process concept tokens sequentially at each decoder layer, similar to Yang et al. (2022).

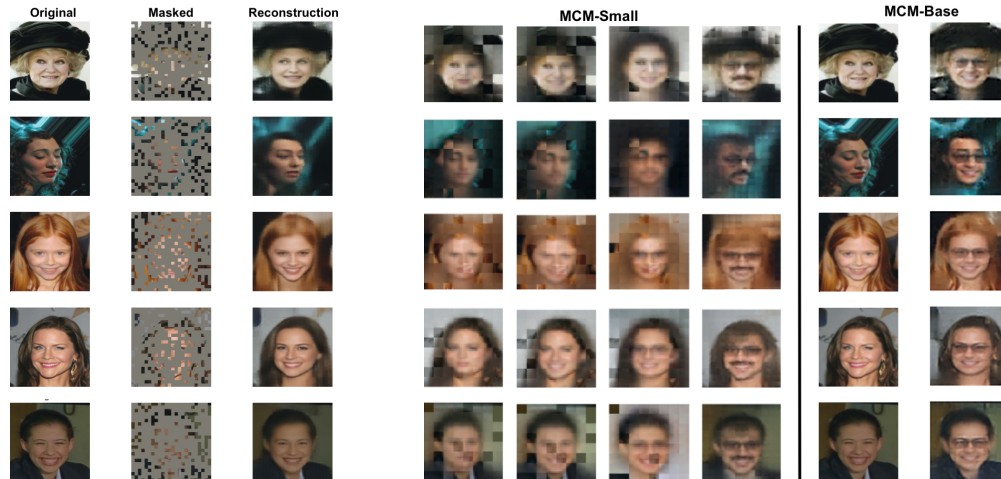

(a) Image reconstruction results for the large-sized model. With masked images as inputs, the model reconstructs plausible complete images from the unmasked patches. Note that the presented images are all test-time results, which precludes the possibility of data memorization.

(b) Image editing results. For MCM-Small, from left to right: 'Not Smiling', 'Male', 'Eyeglasses', and 'Eyeglasses + Male + Bangs + Mustache'. The learned concept tokens exhibit compositional abilities for the reconstruction task. Moreover, for MCM-Base, left: original; right: 'Smiling + Eyeglasses'. We demonstrate enhanced image editing quality when scaling up the model.

Figure 3: Masked image reconstruction and editing results using a high test-time mask ratio of 75%.

(2) **W/O Learnable Latent Concepts:** We replace the learnable concept tokens at each encoder layer with the fixed concept label embeddings $c_{\text{prototype}}$, which query the encoder layer's output via cross-attention. The resulting weighted encoder output is then employed as the input to the corresponding decoder layer, i.e., $\hat{v}_{\text{full}}^{l_{\text{decoder}}} \leftarrow \text{MHA}(v_{\text{full}}^{l_{\text{decoder}}}, \text{MHA}(c_{\text{prototype}}, v_{\text{masked}}^{l_{\text{encoder}}}))$ where $l_{\text{encoder}} = L_{\text{encoder}} - 2l_{\text{decoder}}$.

(3) **Repetitive Latent Concepts:** The learned concept tokens in the latent space are used as the keys and values for cross-attention computations across all decoder layers. In other words, we replace the encoder's layer-wise learned concept embeddings with the same latent concept embeddings.

The weighted concept loss balances gradient assignments for imbalanced concept classes, thus improving recall with enhanced sensitivity to minority classes ('W/ Weighted Loss' ablation in Table 3). The disentanglement loss enhances the decoder's capability to reconstruct masked images from concept tokens, thereby enhancing reconstruction quality with significantly reduced FID scores. Consequently, incorporating both losses resulted in the best concept prediction performance, as evaluated by the test accuracy and F1-score, while maintaining a low FID score of 1.605, effectively balancing image quality. Moreover, the branches architecture plays a significant role in enhancing performance, as its absence led to a sizable decrease in F1-score and an increase in FID ('W/O Branches' ablation). The 'W/O Learnable Latent Concepts' ablation and 'Repetitive Latent Concepts' ablation highlight the efficacy of the proposed concept learning mechanism via cross-attention. While ablating these components was not as impactful as removing the branches themselves, it still resulted in degraded performance for both concept prediction and reconstruction tasks. Consequently, the complete model, with all components included, achieved the best performance.

## 4.3 QUALITATIVE RESULTS

### 4.3.1 MASKED IMAGE RECONSTRUCTION AND EDITING

Masked image reconstruction involves predicting the original image from a partially masked input, where patches are randomly removed. We demonstrate the performance of our approach (training mask ratio 25%) in reconstructing images with a significant portion of patches masked during testing (e.g., 75%) in Figure 3a. Additionally, by providing specific text-based concept tokens, we can traverse the concept latent space and manipulate the reconstruction of the masked patches. The

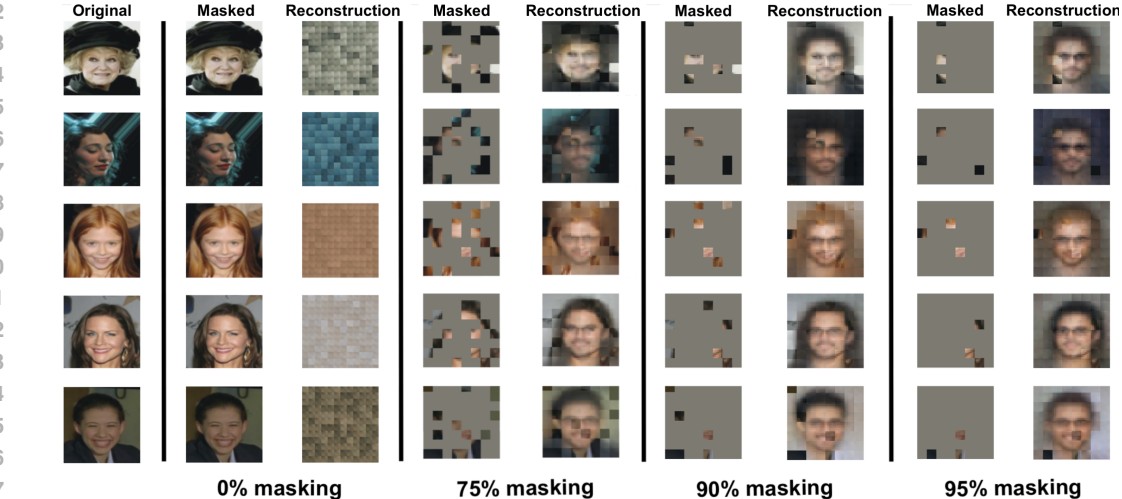

Figure 4: We could employ masks of any arbitrary size during the test phase. A larger mask size (e.g., 95%) provides a reconstruction that better represents the edited concepts, while a smaller mask size (e.g., 0%) generates images that align more closely with the contexts.

resulting image is reconstructed to align with both the contextual unmasked patches and the specified concepts. Notably, we can either activate each concept individually or combine multiple concepts, as illustrated in Figure 3b. We also compare editing results across various model sizes with different computational costs, highlighting that the model's ability to perform masked image editing and reconstruction improves progressively as it scales. In comparison to MCM, methods such as Masked Autoencoder He et al. (2022) lack the ability for image editing while reconstructing the masked image patches. Inducing the concept learning ability in methods such as Masked DiT v2 Gao et al. (2023) can also lead to a substantial degradation in output image quality, as shown in Appendix A.4.

### 4.3.2 VARYING THE TEST MASK RATIO

Figure 4 illustrates how the reconstruction results change with different mask ratios during testing, producing a range of reconstructed images that blend visible contextual patches with the provided concepts, proportional to the chosen ratio.

## 5 CONCLUSIONS

We introduced the Multi-layer Concept Map (MCM) for efficient concept learning from masked images. MCM employs a reconstruction target enhanced by the weighted concept and disentanglement losses, reducing computational cost while maintaining competitive performance in concept learning. MCM enables effective image editing, producing diverse blends of concepts that align with visible contextual patches for the reconstruction task. We hope this work contributes to more efficient concept learning and enhanced interpretability with large-scale Transformer models.

**Limitations.** Unlike conventional concept learning methods that rely on binary labels, MCM utilizes concept embeddings derived through self-supervised learning with the pretrained CLIP model. However, in practice, collecting paired concept-image samples is still necessary for learning effective concept embeddings. Moreover, the concept latent space is not inherently designed for continual learning, where concept classes evolve over time. Future research on dynamically expanding and reusing learned concepts would be valuable for enhancing adaptability in such settings.

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

# A   APPENDIX

## A.1   MODEL SETTINGS

Common parameters across all models (Table 4) include the AdamW optimizer with a learning rate of $1 \times 10^{-3}$, a weight decay of 0.01, four self-attention heads, hidden layer size of 512, and a batch size of 1024. The Small model features 2 encoder layers, 1 decoder layer, an MLP size of 128, and was trained for 500 epochs on 48x48 images with a patch size of 6. The Base model expands to 6 encoder layers, 3 decoder layers, an MLP size of 512, and used 96x96 images with a patch size of 8. Finally, the Large model utilizes 12 encoder layers, 6 decoder layers, an MLP size of 1024, and 192x192 images with the same patch size of 8 but was trained for only 100 epochs due to its increased complexity. The coefficients $\alpha$ and $\beta$ have a default value of one.

## A.2   UNBALANCED CONCEPT CLASS DISTRIBUTION IN CELEBA

The concepts in CelebA exhibit varying frequencies (see Figure 5); for instance, "Mustache" is a rare concept, making it particularly challenging to learn. Consequently, we devise the weighted concept loss to tackle the challenge posed by unbalanced concept classes effectively.

## A.3   ALGORITHM

The complete algorithm is demonstrated in Algorithm 1.

## A.4   COMPARISON OF MASKED IMAGE RECONSTRUCTION QUALITY

We demonstrate in Figure 6 the difference in reconstruction quality of the methods discussed in the experiment section. In particular, we found that although the Masked Autoencoder (MAE) achieves a better FID score, its reconstruction results cannot be edited using the learned concepts. Moreover, methods such as MDTv2 suffer from substantial degradation in output image quality. Our method MCM, however, achieved a balance between reconstruction quality and concept-based image editing ability.

Table 4: Hyperparameters for different model sizes

| Parameter | Value |
|---|---|
| **Common Parameters** | |
| Optimizer | AdamW |
| Weight decay | 0.01 |
| Learning rate | $1 \times 10^{-3}$ |
| Number of self-attention heads | 4 |
| Size of hidden layers | 512 |
| Batch size | 1024 |
| **Small Model Parameters** | |
| Number of encoder layers | 2 |
| Number of decoder layers | 1 |
| Size of MLP | 128 |
| Epochs | 500 |
| Image size (CelebA) | 48 |
| Patch size (CelebA) | 6 |
| Number of parameters | 46.0M |
| **Base Model Parameters** | |
| Number of encoder layers | 6 |
| Number of decoder layers | 3 |
| Size of MLP | 512 |
| Epochs | 500 |
| Image size (CelebA) | 96 |
| Patch size (CelebA) | 8 |
| Number of parameters | 112.2M |
| **Large Model Parameters** | |
| Number of encoder layers | 12 |
| Number of decoder layers | 6 |
| Size of MLP | 1024 |
| Epochs | 100 |
| Image size (CelebA) | 192 |
| Patch size (CelebA) | 8 |
| Number of parameters | 377.3M |

## A.5 DESIGN OF THE TESTING TIME MASK

We compare the editing results between the random mask and the square mask methods in Figure 7. Using a random mask generated an image that aligned with both the visible contextual tokens and the provided concepts. However, a square mask produced unnatural and repetitive editing results across samples, which failed to fit the visible contextual tokens.

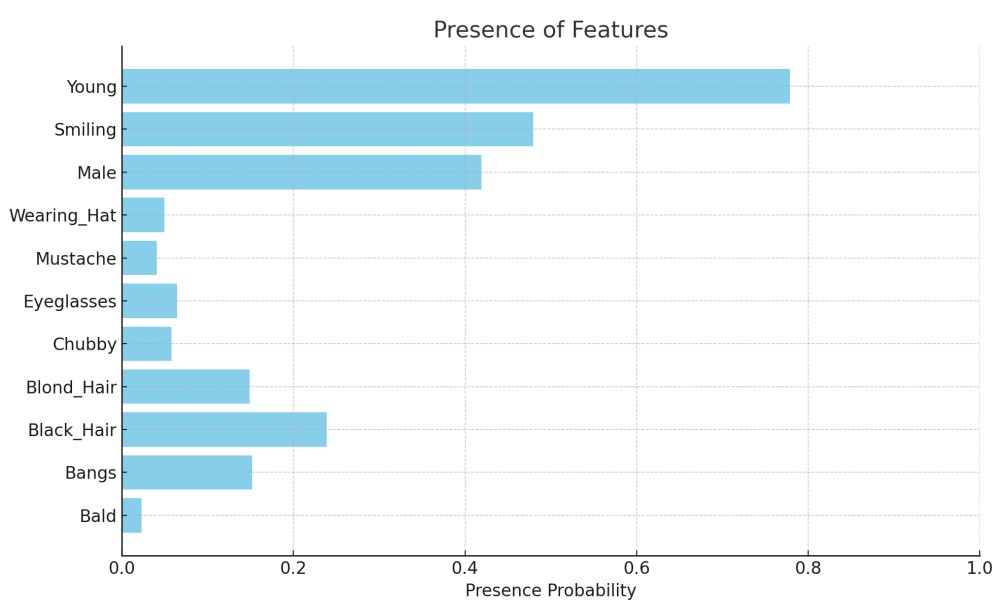

Figure 5: Unbalanced concept classes in the CelebA dataset.

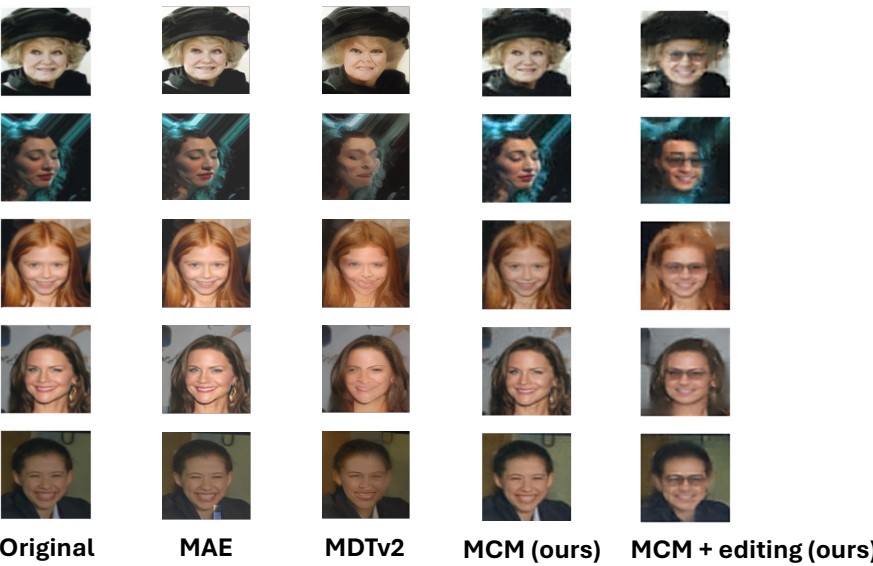

Figure 6: A comparison of image reconstruction quality among different methods learning from masked images.

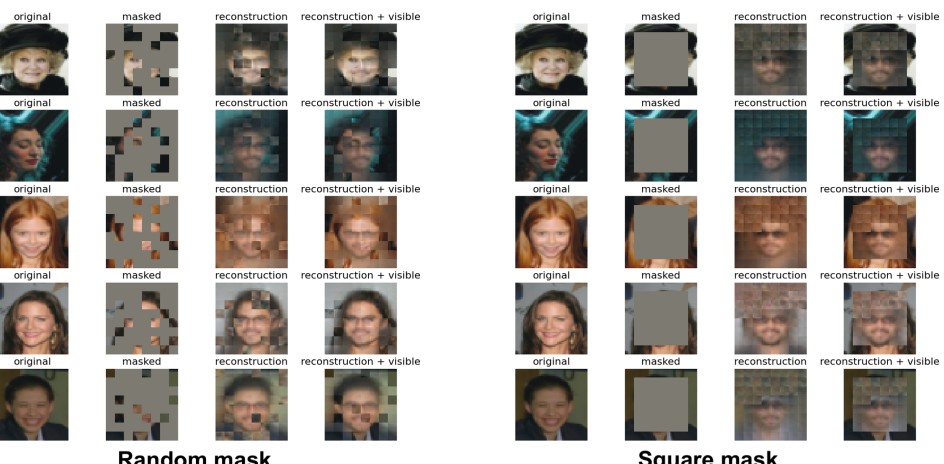

Figure 7: Image editing with different mask shapes based on the concept set 'Male + Eyeglasses + Mustache + Smiling'. The random mask method reconstructs the masked patches while preserving consistency with the specific image contexts. In contrast, the square mask fails to produce diverse results across multiple contexts.

**Algorithm 1** Multi-layer Concept Map (MCM)

1: **Input:** image patches: $X \in \mathbb{R}^{B \times N \times (P^2 \times C)}$; image tokens: $V_p \in \mathbb{R}^{B \times N \times E}$; learnable concept tokens: $C_0 \in \mathbb{R}^{B \times M \times E}$; concept embeddings: $C_{\text{prototype}} \in \mathbb{R}^{B \times M \times E}$.
2: $V_{\text{masked}} \in \mathbb{R}^{B \times (N - \lfloor N \times r \rfloor) \times E} \leftarrow$ Randomly mask input tokens with a ratio $r$
   ▷Mask indices: $Z \in \mathbb{R}^{\lfloor N \times r \rfloor}$.
3: **Encoder**
4: **for** layer $l_{\text{encoder}} = 1, 2, \ldots, L_{\text{encoder}}$ **do**
5: $\quad \hat{C}_{l_{\text{encoder}}} \leftarrow \text{MHA}(C_{l_{\text{encoder}}}, V_{\text{masked}}^{l_{\text{encoder}}}, V_{\text{masked}}^{l_{\text{encoder}}})$ ▷ query, key, value.
6: $\quad C_{l_{\text{encoder}}+1} \leftarrow \text{FF}(\hat{C}_{l_{\text{encoder}}})$
7: $\quad$ **if** Every two encoder layers **then**
8: $\quad\quad$ Add $C_{l_{\text{encoder}}+1}$ to the list of learned concepts $C_{\text{encoder}}\{C_3, C_5, .., C_{l_{\text{encoder}}+1}, ..\}$
9: $\quad$ **end if**
10: $\quad \hat{V}_{\text{masked}}^{l_{\text{encoder}}} \leftarrow \text{SA}(V_{\text{masked}}^{l_{\text{encoder}}})$
11: $\quad V_{\text{masked}}^{l_{\text{encoder}}+1} \leftarrow \text{FF}(\hat{V}_{\text{masked}}^{l_{\text{encoder}}})$
12: **end for**
13: The weighted concept loss: $\ell_{\text{concept}}(C_{L_{\text{encoder}}}, C_{\text{prototype}}) = \frac{1}{B} \sum_{i=1}^{B} \sum_{j=1}^{M} w_{i,j} \cdot \left(c_{L_{\text{encoder}}}^{i,j} - c_{\text{prototype}}^{i,j}\right)^2$.
14: **Decoder**
15: Initialize learnable mask tokens $V_{\text{init}} \in \mathbb{R}^{B \times \lfloor N \times \gamma \rfloor \times E}$ with values drawn from a Gaussian distribution.
16: $V_{\text{full}}^0 \leftarrow$ Concatenate $V_{\text{masked}}^{L_{\text{encoder}}}$ and $V_{\text{init}}$, and rearrange based on the mask indices $Z \in \mathbb{R}^{\lfloor N \times \gamma \rfloor}$.
17: **for** layer $l_{\text{decoder}} = 1, 2, \ldots, L_{\text{decoder}}$ **do**
18: $\quad$ Retrieve learned concepts from the encoder $C_{l_{\text{decoder}}} \leftarrow C_{\text{encoder}}[-l_{\text{decoder}}]$.
19: $\quad \hat{V}_{\text{full}}^{l_{\text{decoder}}} \leftarrow \text{MHA}(V_{\text{full}}^{l_{\text{decoder}}}, C_{l_{\text{decoder}}}, C_{l_{\text{decoder}}})$.
20: $\quad V_{\text{full}}^{l_{\text{decoder}}+1} \leftarrow \text{FF}(\hat{V}_{\text{full}}^{l_{\text{decoder}}})$
21: **end for**
22: Convert $V_{\text{full}}^{L_{\text{decoder}}}$ into pixel level images $\hat{X} \in \mathbb{R}^{B \times N \times (P^2 \times C)}$
23: Compute the masked reconstruction loss: $\ell_{\text{re}}(X, C_0) = \frac{\frac{1}{B} \sum_{j=1}^{N} \sum_{i=1}^{B} (\hat{x}_{i,j} - x_{i,j})^2 \cdot \mathbb{1}[j \in Z]}{\lfloor N \times \gamma \rfloor}$.
24:
25: **Disentanglement Loss**
26: Select a specific concept position $j \in \{1, 2, ..., M\}$ based on a random binary mask $U \in \{0, 1\}^M$.
27: Replace the concept $C_{L_{\text{encoder}}}^j$ with its antonym token $\mathcal{O}(C_{L_{\text{encoder}}}^j)$: $\hat{c}_{L_{\text{encoder}}}^i = \{U_j \cdot \mathcal{O}(c_{L_{\text{encoder}}}^{i,j}) + (1 - U_j) \cdot c_{L_{\text{encoder}}}^{i,j}\}_{j=1}^M$.
28: Reconstruct an image based on the modified concepts: $\tilde{x}_i \leftarrow f_{\text{decoder}}(\hat{c}_{L_{\text{encoder}}}^i)$.
29: Predict the concepts in $\tilde{x}_i$ using the encoder: $\tilde{c}_{L_{\text{encoder}}}^i \leftarrow f_{\text{encoder}}(\tilde{x}_i)$.
30: Compute the disentanglement loss: $\ell_{\text{disentangle}}(X, C_0, U) = \frac{1}{B} \sum_{i=1}^{B} (\hat{c}_{L_{\text{encoder}}}^i - \tilde{c}_{L_{\text{encoder}}}^i)^2$.
31: The final loss: $\mathcal{L}(X, C_0, C_{\text{prototype}}, U) = \ell_{\text{re}}(X, C_0) + \alpha \cdot \ell_{\text{disentangle}}(X, C_0, U) + \beta \cdot \ell_{\text{concept}}(X, C_0, C_{\text{prototype}})$
   ▷$\alpha, \beta$ : coefficients.

