# OpenReview forum: "MCM: Multi-layer Concept Map for Efficient Concept Learning from Masked Images"
_ICLR.cc/2026/Conference — ICLR 2026 Conference Withdrawn Submission_

### Official Review · Reviewer_zFEV · 2025-10-28

**Soundness:** 2
**Presentation:** 3
**Contribution:** 1
**Rating:** 2
**Confidence:** 5

**Summary:**

This work proposes Multi-layer Concept Map (MCM), a vision Transformer framework that combines masked image reconstruction with concept learning. The core idea is an asymmetric encoder–decoder architecture where learnable concept tokens are processed alongside unmasked tokens through multiple encoder layers, and these concept tokens are then used via cross-attention in several decoder layers.

**Strengths:**

MCM is demonstrated on the CelebA dataset for predicting face attribute concepts and reconstructing masked images, which enables image editing by manipulating concept tokens. The authors highlight architectural novelty and improved efficiency as the main contributions.

**Weaknesses:**

The paper suffers from several critical weaknesses in novelty, experimental validation, and clarity of motivation, as detailed below.

1. The architectural contributions are not truly novel. The asymmetric encoder–decoder with a lightweight decoder for masked image modeling is directly inspired by MAE; thus, MCM’s use of a mask-based asymmetric architecture is an application of known methods rather than a new invention. Further, Introducing learnable concept tokens is positioned as a key novelty, but this closely parallels ideas from concept bottleneck models and visual concept tokenization. Similarly, the proposed loss functions, the disentanglement loss and weighted concept loss, provide only incremental improvement and are standard techniques in concept learning, rather than innovative algorithms.

2. All experiments are performed only on the CelebA dataset, and further, only 11 concepts (out of 40 attributes) are used.
This raises serious concerns regarding generalisation to other datasets, scalability with respect to the number of concepts, and overall practicality.

The decision to focus on 11 “easy” concepts is understandable for stability, but it likely inflates apparent performance. Harder or subtler concepts might reveal that MCM’s tokens do not generalise well. Moreover, without a principled selection criterion, this subset appears hand-tuned.

It would strengthen the paper to report results on all 40 CelebA attributes, even if weaker, to demonstrate generalisation. Also, to evaluate on additional datasets (e.g. LFWA) with non-overlapping concept sets.

3. The authors claim MCM is “efficient”; however, the model uses only 25% masking, which modestly reduces computation at best.
The introduction of concept tokens likely offsets these savings, and there is no analysis of training/inference time or memory usage to support the efficiency claim. Without such evidence, the claim of efficiency is unconvincing.

4. The sharp performance drop of ViT-Large at a 0.1 masking ratio is unjustified. If this is due to random initialisation sensitivity, that should be demonstrated with multiple seeds. Moreover, since the 0% (no mask) case achieves almost the same performance as the 0.25 mask, the masking strategy appears to provide only marginal benefit, calling into question whether it is essential to the proposed method at all.

5. Table 2, which provides the only comparison to SOTA, represents an unfair setup. The baseline MAE + MLP uses a simple classifier on top of unmasked tokens representations, where the loss is overwhelmed by pixel reconstruction; the concept head receives tiny gradients that do not shape encoder features. In contrast, MCM injects concept supervision deeply throughout the model and benefits from stronger signals. Therefore, the comparison does not isolate architectural improvements, it conflates different supervision regimes.

6. The visualisation of 0% masking in Figure 4 is counterintuitive. One would expect that more visible context leads to better reconstruction especially that the model is trained with only 25% masking. However, the model fails in this case, producing averaged outputs.

7. The authors list “novel image editing capabilities for masked image reconstruction” as a main strength, stating that this is a functionality MAE cannot provide. However, there is no evidence supporting this claim. Even in Figure 6, the “edited” images look almost identical, same smiles, cheeks, glasses, and overall appearance. The model does not demonstrate any genuine or personalised semantic edits, calling into question this claimed advantage.

-----------

Beyond technical implementation, the purpose of the paper remains unclear. While the method combines MAE and concept learning, the authors never articulate a convincing reason why such a hybrid is necessary or beneficial. If the goal is concept-controllable image editing, there are stronger and more established models.

**Questions:**

Addressing the mentioned limitations would significantly improve the clarity, credibility, and impact of the paper. However, the main concerns with this work relate to its overall novelty, motivation, and experimental scope, rather than clarifications that could be addressed in a short rebuttal. These issues appear more structural than technical and would likely require substantial rethinking rather than minor revisions.

---

### Official Review · Reviewer_eBrH · 2025-10-28

**Soundness:** 3
**Presentation:** 3
**Contribution:** 3
**Rating:** 4
**Confidence:** 3

**Summary:**

The paper proposes Multi-layer Concept Map (MCM), an architecture for learning human-interpretable visual concepts directly from masked images, and using those learned concepts to guide reconstruction of the missing regions. MCM first encodes only the visible patches of an input image along with a small set of learnable “concept tokens”; at each encoder layer, cross-attention updates these concept tokens so that different layers capture concepts at different levels of granularity, without using self-attention between concept tokens to keep them disentangled. The decoder then reconstructs the masked patches by attending to these layer-specific concept tokens (rather than just visible context), using an asymmetric design where every two encoder layers supervise one decoder layer to reduce compute. Training of MCM composes three loss terms: a masked reconstruction loss, a disentanglement loss that enforces that each concept token controls a distinct semantic factor, and a weighted concept loss that upweights rare concepts using frequency-based weights. The method is evaluated on CelebA for both concept prediction (accuracy, precision/recall/F1) and reconstruction quality (FID)。

**Strengths:**

1. This paper has a solid motivation. The author tackles concept learning from masked images, which is underexplored, and proposes the MCM as an explicit solution.
2. MCM benefits from asymmetric encoder-decoder design, similar to MAE, where only the visible patches are passed through the encoder. This design promotes efficiency.
3. The work evaluates across multiple MCM sizes and performs comprehensive ablations that isolate each proposed component.
4. Empirical results show good tradeoff in training time and performances.

**Weaknesses:**

1. All experiments are on single dataset CelebA with only 11 attributes. Without empirical results on dataset with other attributes, this limits the generalization beyond faces or to richer concept taxonomies.
2. Disentanglement loss and weighted concept loss are strictly tied up to the predefined list of concepts. This limits the continual learning or expanding the concept set.
3. The author included the disentanglement loss to forcibly disable self-attention among concept tokens, which can hinder modeling dependencies (co-occurrence, mutual exclusivity) between concepts and limit compositional reasoning. The author should provide more analysis on the cons and pros for including this disentanglement loss.
4. The direct application of this MCM is unclear. While the qualitative examples present in Figure 3 and 4 show the potential of image editing, it is unclear to me how MCM goes beyond prediction of finite facial attribute. The author should consider include broader implication of MCM and provide more insights into how this work will be beneficial.

**Questions:**

1. I'm wondering to what extent the concept disentanglement is useful and beneficial.

---

### Official Review · Reviewer_8PL8 · 2025-10-30

**Soundness:** 2
**Presentation:** 3
**Contribution:** 2
**Rating:** 2
**Confidence:** 4

**Summary:**

This paper introduces Multi-layer Concept Map (MCM) for Concept learning. The goal is to learn concept representations that can be used both for classification and for concept-driven counterfactual image reconstruction. The motivations of the method are to make concept learning more efficient via masking image parts in the encoder. MCM is a encoder-decoder architecture, where a ViT encoder transforms image and learnable concept tokens. Image tokens are transformed via regular Self-attention blocks, and then used for keys and values in cross-attention which transforms the concept tokens.
The decoder ingests the encoded image tokens, as well as an appropriate number of mask tokens, and cross-attends to concept tokens. The model is trained via a mixture of three objectives:
* decoder reconstruction loss
* concept loss – forces the encoder outputs of the concept tokens to mimic CLIP text encoder embeddings of the concept names - potentially reweighted to account for the sparsity of certain concepts.
* disentanglement loss – we replace random concept embeddings in the encoder output with their antonyms (i.e. embeddings of the antonyms of concept names), feed them through the decoder, feed the output through the encoder, and force the similarity of the encoder outputs to the antonyms.

The model exhibits the ability to accurately recognize the learned concepts, as well as generate counterfactual images with the reversed concepts.

**Strengths:**

1. MCM is an interesting way to generate counterfactual predictions.
2. The proposed method learns strong concept representations.
3. MCM enables flexible test-time control over edit strength via mask ratio, which is an appealing property.

**Weaknesses:**

1. The architecture largely resembles prior cross-attention masked autoencoders; the main semantic capability is imported from CLIP embeddings rather than emerging from reconstruction. As such, the methodological novelty appears incremental.
2. While MCM “does not require binary concept labels for training” (L358), it uses CLIP embeddings as targets derived from those exact binary labels - an almost equivalent form of supervision.
3. The experimental section is limited to only the CELEB-A dataset. It remains unclear whether MCM scales to domains with richer compositional structure. The experiments would be more compelling if they included other datasets and concepts, for example the CUB dataset and prototypes learned by Prototypical Part Networks [1].
4. Despite being one of the core contributions of the method, the counterfactual generation ability is not evaluated quantitatively (e.g. by reporting the efficacy of fooling the encoder with switched concepts). Leaning into this property of the model could potentially strengthen the contributions of the paper.
5. Evaluation details (e.g., mask sampling, reconstruction metrics, classifier design) are insufficiently described for reproducibility, and no code is provided.
6. CLIP is described as self-supervised, which is not accurate; CLIP uses natural-language supervision.

[1] This Looks Like That: Deep Learning for Interpretable Image Recognition
Chaofan Chen, Oscar Li, Chaofan Tao, Alina Jade Barnett, Jonathan Su, Cynthia Rudin https://arxiv.org/abs/1806.10574

**Questions:**

1. Per Table 1, the increased mask ratio leads to only marginal reduction in training time (e.g. from 9.8 to 8.1 hours). Previous literature (e.g. MAE [2]) reports three-fold speedup thanks to masking. Why is that the case?
2. What would be the result of using binary concepts as targets and decoder inputs, instead of CLIP embeddings?
3. Can we see a comparison of images generated by MCM and other approaches which report low FID, especially the MAE?


[2] Masked Autoencoders Are Scalable Vision Learners
Kaiming He, Xinlei Chen, Saining Xie, Yanghao Li, Piotr Dollár, Ross Girshick
https://arxiv.org/abs/2111.06377

---

### Official Review · Reviewer_uiqj · 2025-10-30

**Soundness:** 2
**Presentation:** 3
**Contribution:** 2
**Rating:** 4
**Confidence:** 3

**Summary:**

In this paper, the authors propose a novel concept learning framework that leverages masked images to learn a set of concept tokens. These tokens are randomly initialized and then optimized through a multi-layer concept decoder, which performs image reconstruction guided by the learned concepts. To enhance interpretability, a disentanglement loss is introduced to ensure that each concept token controls a distinct semantic aspect, while a weighted concept loss is employed to address the challenge of unbalanced concept distributions. Experiments on the CelebA dataset demonstrate that the learned concept tokens enable both accurate concept classification and controlled image reconstruction, outperforming existing baselines.

**Strengths:**

- The paper is well written and easy to follow.
- The model trained on CelebA show both quantitative and qualitative improvements over baselines.
- The asymmetric architecture make the training to be efficient.

**Weaknesses:**

- The novelty is somewhat incremental, as it mainly integrates known components  into a single framework.
- The method is only evaluated on CelebA, which is a relatively simple and small dataset; it is  unclear whether the approach generalizes to more complex or non-face domains.
- The model will need the pretrained CLIP to get the concept embeddings, so it is somehow like distililling the knowledge but not acctually the proposed method's effect.
- According to the ablation the proposed looses, the proposed losses improve the performance quite marginal.

**Questions:**

- The model appears to be quite sensitive to the mask ratio, particularly for larger models. It would be helpful to discuss whether there exists a more principled way to determine the optimal mask ratio beyond grid search, especially when applying the method to large-scale datasets where exhaustive tuning becomes impractical.

---

### Note · Authors · 2025-11-13

I have read and agree with the venue's withdrawal policy on behalf of myself and my co-authors.